# In Vitro Digestion and Colonic Fermentation of UHT Treated Faba Protein Emulsions: Effects of Enzymatic Hydrolysis and Thermal Processing on Proteins and Phenolics

**DOI:** 10.3390/nu15010089

**Published:** 2022-12-24

**Authors:** Jingyu Gu, Minhao Li, Malik Adil Nawaz, Regine Stockmann, Roman Buckow, Hafiz A. R. Suleria

**Affiliations:** 1School of Agriculture and Food, Faculty of Veterinary and Agricultural Sciences, The University of Melbourne, Parkville, VIC 3010, Australia; 2Commonwealth Scientific and Industrial Research Organisation (CSIRO), Agriculture and Food, Werribee, VIC 3030, Australia; 3Centre for Advanced Food Engineering, School of Chemical and Biomolecular Engineering, The University of Sydney, Darlington, NSW 2006, Australia

**Keywords:** faba proteins, enzymatic hydrolysis, UHT processing, emulsion, phenolic compounds, in vitro digestion, colonic fermentation

## Abstract

Faba bean (*Vicia faba* L.) protein is a new plant protein alternative source with high nutrient content especially protein and phenolic compounds. The present study investigated physicochemical properties, phenolic content, antioxidant potential, and short chain fatty acids (SCFAs) production during in vitro digestion and colonic fermentation of faba bean hydrolysates and oil-in-water (O/W) emulsions. Results indicate that the enzymic hydrolysates of faba proteins exhibited higher protein solubility, increased electronegativity, and decreased surface hydrophobicity than native faba protein. O/W emulsions showed improved colloidal stability for the faba protein hydrolysates after ultra-high temperature processing (UHT). Furthermore, UHT processing preserved total phenolic content, DPPH and ABTS radical scavenging abilities while decreasing total flavonoid content and ferric reducing power. Besides, the release of phenolic compounds in faba bean hydrolysates (FBH) and emulsions (FBE) improved after intestinal digestion by 0.44 mg GAE/g and 0.55 mg GAE/g, respectively. For colonic fermentation, FBH demonstrated an approximately 10 mg TE/g higher ABTS value than FBE (106.45 mg TE/g). Total SCFAs production of both FBH and FBE was only 0.03 mM. The treatment of FBH with 30 min enzymatic hydrolysis displayed relatively higher antioxidant capacities and SCFAs production, indicating its potential to bring more benefits to gut health. Overall, this study showed that enzymic hydrolysis of faba proteins not only improved the colloidal emulsion stability, but also released antioxidant capacity during in vitro digestibility and colonic fermentation. Colonic fermentation metabolites (SCFAs) were related to the degree of hydrolysis for both FBH and FBE. Additional studies are required to further elucidate and differentiate the role of phenolics during faba protein processing and digestion stages in comparison to contributions of peptides, amino acids and microelements to digestion rates, antioxidant capacities and colonial SCFA production.

## 1. Introduction

The Faba bean (*Vicia faba* L.) is one of the most widely grown winter season legume crops [1]. As a member of the Fabaceae family, it is typically named fava bean or broad bean, and usually cultivated as food and fodder [2,3,4]. Like other legumes, faba bean is a good source of quality protein (27–34%) [5] and phenolics, especially condensed tannins [6]. In addition, a high proportion of other minor compounds are also present in faba beans, such as levodopa as well as phytic acids which may aid in treating Parkinson’s disease and have anti-cancer properties [7,8]. Recently, faba protein along with other legume proteins has gained popularity due to lower environmental impacts and lower cost of production compared to animal proteins [9]. However, application of faba proteins in plant based foods has not yet been fully explored and faba proteins are still considered an emerging legume protein source.

Phenolic-protein interactions may result in nutrition losses due to protein precipitation and the inactivation of digestive enzymes [10]. Moreover, complexes between phenolics and proteins may affect the bio-accessibility and bioavailability of proteins and phenolics by protecting them from oxidizing as they pass through the GI tract [11]. To improve the bioavailability of phenolics and protein digestibility, protein hydrolysis may be commercially implemented by using protease enzymes such as Alcalase to help decompose protein into smaller peptides and free amino acids, thereby promoting antioxidant capacity [12,13]. Several processing techniques (such as ultra-high temperature processing, high shear strength mixing, and homogenization) also assist in enhancing the stability of protein emulsions by generating pressure and heating impacts [14], which further provide specific antioxidant potential due to the formation of some Maillard reaction products [15].

Earlier studies reported that protein hydrolysates derived from Alcalase showed more resistance to digestive enzymes and higher antioxidant activities than hydrolysates generated from other proteases [16,17]. However, to the best of our knowledge, the influence of Alcalase hydrolysis on phenolic compounds contents and activity in hydrolysates of faba bean is not understood. Furthermore, the knowledge on how UHT, an important commercial processing operation to stabilize plant-based drinks, transforms the phenolic contents and antioxidant properties of faba bean protein emulsions is limited. Lastly, there is a lack of comprehensive understanding of how in vitro gastrointestinal digestion influences the release of phenolic compounds in hydrolysates and emulsions. Therefore, the objective of the present study was to determine how physicochemical properties, phenolics contents, and antioxidant activities, changed during in vitro gastrointestinal digestion and colonic fermentation of oil-in-water (O/W) emulsions formed with functional faba been protein hydrolysates with subsequent UHT processing treatment. The production of SCFAs, of FBH and FBE that is generated during colonic fermentation in a faecal model, was also examined.

## 2. Materials and Methods

### 2.1. Chemicals and Reagents

All chemicals and reagents used were of analytical grade and purchased from Sigma-Aldrich (St. Louis, MO, USA) unless otherwise stated.

### 2.2. Sample Preparation

Commercial dehulled faba beans were locally purchased, ground into powders and defatted with n-hexane and stored at room temperature (Experimental design of the study is presented in Figure 1).

#### 2.2.1. Extraction of Faba Bean Phenolics (FBP)

Ethanolic extraction of faba bean was performed according to the previously published method with slight modifications [18]. Briefly, slurry of faba bean powder (5 g) was prepared in 20 mL ethanol (70% *v*/*v*) by homogenising with Ultra-Turrax T25 Homogenizer (IKA Werke GmbH & Co. KG, Staufen, Germany) for 30 s at 10,000 rpm. Homogenised samples were incubated in a shaking incubator (ZWYR-240 incubator shaker, Labwit, Ashwood, VIC, Australia) for 12 h at 4 °C for 120 rpm. Subsequently, incubated samples were centrifuged by Hettich Refrigerated Centrifuge (ROTINA380R, Tuttlingen, Germany) at 5000 rpm for 15 min at 4 °C.

#### 2.2.2. Preparation of Faba Bean Protein Concentrate (FBPC)

Faba bean flour was defatted using n-hexane. Defatted flour was mixed with distilled water (1:10) to make a slurry and pH was adjusted to 9.0 using 2 N NaOH. The slurry was then mixed for 2 h at room temperature using an overhead stirrer, followed by centrifugation at 10,000× *g* for 15 min at 10 °C and collection of the supernatant. The pH of the supernatant was adjusted to 4.3 and centrifuged at 10,000× *g* for 15 min 10 °C. The pellets were separated from the supernatant, and pH of the pellets was adjusted to 7.0, followed by freeze drying. The composition and caloric value of the concentrate (per 100 g) were carbohydrate (27 g), total fat (3 g), protein content (63 g), total ash (4 g), moisture content (3 g), and calculated calories (410 kcal).

#### 2.2.3. Preparation of Faba Bean Hydrolysates (FBH)

FBPC was hydrolysed by Alcalase according to the method described by Ghribi et al. [19]. Briefly, 10% (*w*/*w*) FBPC dispersions pre-equilibrated at pH 7.0 were hydrolysed using 0.5% (*v*/*v*) of Alcalase (2.4 AU/g of protein) (Alcalase^®^, Novozymes Australia Pty. Ltd., North Ride, NSW, Australia) at 50 °C for various time periods viz., 5 min (A₅), 10 min (A₁₀), 15 min (A₁₅), and 30 min (A₃₀). After the target time of reaction, hydrolysis was stopped by inactivating Alcalase at 85 °C for 20 min. Control sample (slurry without pH adjustment and Alcalase) and A₀ sample (slurry with adjusted pH 8 and temperature treatment without Alcalase) were also prepared.

#### 2.2.4. Preparation of Faba Bean Emulsions (FBE)

O/W emulsions (approximately 3000 g) were prepared by adding 5% (*w*/*w*) of native FBPC (C) or hydrolysed FBPC (A₀, A₅, A₁₀, A₁₅, and A₃₀), 1% (*w*/*w*) sunflower oil and 0.2% (*w*/*w*) sunflower lecithin (as a co-emulsifier) in Milli-Q grade water (pH 7.0). The dispersions were mixed with a multi-mixer at 1000 rpm for 10 min. Following mixing, the coarse emulsions were prepared by homogenising the lipidic and aqueous phases using a Silverson high-shear laboratory mixer (Silverson^®^, Artarmon, NSW, Australia) at 10,000 rpm for 20 min. The coarse emulsions were homogenised by two passes in an EmulsiFlex-C5 (Avestin, Ottawa, Canada) operating at ~80 MPa followed by UHT processing using a bench-top UHT unit (FT74XTS: UHT/HTST System, Armfield Ltd., Hampshire, UK), pre-heated to 105 °C for 3 s and operated at high heat-treatment at 135 °C for 3 s with a sample flow rate of 3 mL/s.

### 2.3. In Vitro Gastrointestinal Digestion

FBH and FBE were digested in three stages, including oral, gastric and intestinal, followed by the harmonized INFOGEST 2.0 protocol of static in vitro GID conditions, including enzymes, CaCl_2_ and the simulated oral (SOF), gastric (SGF) and intestinal (SIF) fluids described by Sánchez-Velázquez et al. [20] with some modifications. All the samples were first stirred in water (1:2, *w*/*v*) and an aliquot of 5 mL was taken as a non-digested (ND) aliquot. Samples were then dissolved in SOF (1:1, *v*/*v*) at pH 7.0 to add 75 U of α-amylase/mL and stirred at 37 °C for 2 min. 5 mL was then taken as an oral-phase aliquot. To simulate the gastric phase (GP), oral-phase bolus was mixed with SGF (1:1, *v*/*v*) at 37 °C and adjusted to pH 3 before adding 2000 U of porcine pepsin/mL. Gastric phase (GP) aliquots were removed after incubation at 37 °C for 2 h. Gastric-digested samples were further digested by dissolving in SIF (1:1, *v*/*v*), adjusted pH to 7.0, trypsin (100 U/mL), and bile salt (10 mM) were added to the intestinal phase (IP). Intestinal phase aliquots were taken after incubation at 37 °C for 2 h.

### 2.4. In Vitro Colonic Fermentation

The in vitro colonic fermentation procedure was conducted by using a modified method of Gu et al. [21]. A pig faecal model considered a good predictor for human colonic fermentation was selected. Briefly, Pig faecal samples were used as gut microbiome source as human faeces substitutes since pigs and humans are primarily colonic fermenters sharing a comparable gut microbiome. Ten mixed male and female large landrace grower white pigs (around 50 kg live weight) and raised in animal house of Diamond Valley Pork (Laverton North, VIC, Australia), with the standard grower diets for two weeks. The faeces were taken immediately after the pigs defecated and were put into an anaerobic chamber. The medium (20% faeces) was prepared by mixing 20 g faeces with 80 g 0.1 M sterilized pre-nitrogen flushed phosphate buffer (pH 7.0) in a stomacher mixer (MiniMix^®^ Lab Blender, Thomas Scientific, Swedesboro, NJ, USA) for 5 min followed by the filtration through a sterile muslin cloth. Sediments from the small intestinal digestion (FBH and FBE) were prepared after centrifugation at 10,000× *g* for 10 min. Then, aliquots of 5 mL faecal slurry were added into six sets of tubes with the sediments and 5 mL of the basal media were added. These six sets of tubes were first flushed with nitrogen and then incubated with shaking at 120 rpm for 0 h, 2 h, 4 h, 8 h, 16 h, and 24 h, respectively in darkness. Afterwards, the tubes were centrifuged at 10,000× *g* and 5 °C for 10 min and the supernatant was taken from the sediment. Supernatants were stored at −80 °C for analysis of phytochemical bioactivity and short chain fatty acid analysis (SCFA) production.

### 2.5. Physicochemical Properties of Hydrolysates and Emulsions

#### 2.5.1. Degree of Hydrolysis (DH)

DH of all FBE was calculated by quantifying of α-amino groups released during hydrolysis by o-phthaldialdehyde (OPA) as described by Church et al. [22]. Briefly, on a 96-well fluorescence plate, 5 µL of sample (glycine standards or hydrolysate) was mixed with 200 µL of OPA reagent (97.5 mL of 100 mM sodium teteraborate (pH 9.9), 0.5 mL of 20% SDS, 1 mL of 5 mg/mL methylated OPA, 1 mL of 5 mg/mL aqueous dithiothreitol) for 5 min at 25 °C, followed by incubation at 37 °C for 20 min. Fluorescence emission (excitation λ 340 nm, emission λ 450 nm) of incubated samples was measured by a plate reader (Varioskan Flash, Thermo Scientific, Waltham, MA, USA). DH was calculated by using Equation (1):(1)$$DH\;\left(\%\right)=\frac{C_{S}\times D_{F1}\times D_{F2}\times 100}{m_{3}\times H_{tot}}$$
where, *C_S_* (concentration of sample (mmol/L)), *D_F_*_1_ (dilution factor prior to OPA reaction), *D_F_*_2_ (dilution factor during OPA reaction), *m*_3_ (mass of protein per litre), *H_tot_* (total peptide bonds in protein substrate (assumed as 7.8 mmol/g)).

#### 2.5.2. Protein Solubility

Hydrolysate suspensions (1% *w*/*v*) in 5 mM potassium phosphate buffer (pH 8, prepared in Milli Q water) were prepared by mixing for 10 min. Suspensions were then centrifuged (12,000× *g*, 20 °C for 20 min), followed by estimation of the total nitrogen contents of supernatant using Dumas combustion method with LECO TrumacẀ^®^ N (LECO Corporation, St. Joseph, MA, USA). Total protein contents in the samples were calculated using a conversion factor of 6.25 [23]. The protein solubility was expressed as a percentage of supernatant protein over total protein.

#### 2.5.3. ζ-Potential

The ζ-potential of prediluted (~10-folds) samples FBH and FBE samples were measured using Malvern Zetasizer at 25 °C at a refractive index of 1.33 [24].

#### 2.5.4. Surface Hydrophobicity (S₀)

The S₀ of FBH and FBE were measured using 8-anilino-1-naphthalene sulfonic acid (ANS) as a fluorescent probe on a spectrofluorometer as described by Nwachukwu and Aluko [25]. Briefly, four different serial dilutions (0.006, 0.003, 0.0012 and 0.0009%) of every sample were prepared followed by sub-division of each dilution into two batches (with and without ANS solution). 20 μL of ANS (8.0 mM prepared in 5 mM phosphate buffer) was added to 5 mL of the first batch while the second batch was used as a blank. The ANS-protein conjugation was measured at 370 nm (excitation) and 490 nm (emission) wavelengths using a plate reader (Varioskan Flash, Thermo Scientific, Waltham, MA, USA). The S₀ index was obtained through linear regression analysis of the plot between fluorescence intensity (FI) and protein concentrations (as the slope of fluorescence intensity-protein concentration).

#### 2.5.5. Particle Size

The average particle size distribution of all FBE was determined using a Malvern Mastersizer 3000 (Malvern Instruments Ltd., Worcestershire, UK). The continuous phase was Milli-Q grade water (Refractive Index: 1.33), while the dispersed phase was sunflower oil and plant protein. Measurements for particle size was obtained using ~12.5% laser obscuration and the droplet size distribution proportion at 10% is d(0.1), 50% is d(0.5) and 90% is d(0.9), respectively. The uniformity, surface weighted mean, D^3,2^ and volume-weighted mean D^4,3^ were also achieved from the particle size distribution graphs [26]. Then, the polydispersity index (PDI) was also calculated according to Equation (2).
PDI = d(0.9) − d(0.1)/d(0.5)(2)

### 2.6. Estimation of Phenolics and Their Antioxidant Potential

Estimations were performed according to the methods of Akhtar, Wu, Ponnampalam, Cottrell, Dunshea and Suleria [18] using Multiskan Go microplate photometer (Thermo Fisher Scientific Inc., Waltham, MA, USA).

#### 2.6.1. Determination of Total Phenolic Content (TPC)

Folin–Ciocalteu’s method was used to estimate the TPC of FB, FBPC, FBH, FBE and their corresponding digests. 25 µL extract, 25 µL Folin–Ciocalteu’s reagent solution (1:3 diluted with water) and 200 µL water were added into the 96-well plate (Costar, Corning, NY, USA) and incubated for 5 min at room temperature (25 °C). Afterwards, 25 µL of 10% (*w*/*w*) sodium carbonate was added and incubated for 60 min at 25 °C. Absorbance was then measured at 765 nm using a spectrophotometer (Thermo Fisher Scientific, Waltham, MA, USA). Gallic acid standard curve with concentrations ranging from 0 to 200 µg/mL was used to determine TPC content and expressed in mg of gallic acid equivalents per gram of extract (mg GAE/g).

#### 2.6.2. Determination of Total Flavonoid Content (TFC)

TFC of FB, FBPC, FBH, and FBE and their corresponding digests were quantified by using aluminium chloride. Extract (80 µL), 80 µL of 2% aluminium chloride and 120 µL of 50 g/L sodium acetate solution were added into the 96-well plate and incubated in dark for 2.5 h at room temperature. Absorbance was measured at 440 nm. Quercetin standard curve with concentrations ranging from 0–50 µg/mL was used to determine TFC and expressed in mg quercetin equivalents per gram of extract (mg QE/g).

#### 2.6.3. Determination of Total Condensed Tannin (TCT)

TCT of FB, FBPC, FBH, and FBE and their corresponding digests were measured using vanillin sulphuric acid method. 25 µL of 32% sulphuric acid, 25 µL of sample extract and followed by150 µL of 4% vanillin solution were added to 96-well plate which was covered and incubated for 15 min. The absorbance was measured at 500 nm. Catechin standard curve with concentrations ranging from 0 to 1 mg/mL was used for estimation of TCT and expressed in mg catechin equivalents (CE) per gram of extract (mg CE/g).

#### 2.6.4. 2,2′-Diphenyl-1-Picrylhydrazyl (DPPH) Assay

DPPH (4 mg) was dissolved in methanol (100 mL) to prepare DPPH radical solution. 40 µL of extract and 260 µL of DPPH solution were first added to 96-well plate and then vigorously shaken in the dark for 30 min at 25 °C. The absorbance was measured at 517 nm. A Trolox standard curve with concentrations ranging from 0 to 200 µg/mL was used to evaluate the DPPH radical scavenging activity and expressed in mg of Trolox equivalent per gram (mg TE/g) of extract.

#### 2.6.5. Ferric Reducing Antioxidant Power (FRAP) Assay

Sodium acetate solution (300 mM), TPTZ solution (10 mM) and Fe [III] solution (20 mM) were mixed at the ratio of 10:1:1, to prepare the FRAP solution. Then, 20 µL of the extract and 280 µL prepared dye solution were added to a 96-well plate and incubated for 10 min at 37 °C. The absorbance was measured at 593 nm. Trolox standard curve with concentrations ranging from 0 to 200 µg/mL was used to evaluate the FRAP values and expressed in mg of Trolox equivalent per gram of extract (mg TE/g).

#### 2.6.6. 2,2′-Azino-Bis-3-Ethylbenzothiazoline-6-Sulfonic Acid (ABTS) Assay

ABTS^+^ stock solution was prepared by mixing 5 mL of 7 mM ABTS solution and 88 µL of 140 mM potassium persulfate. Then, the reaction mixture was incubated in dark for 16 h. 10 µL of the extract and 290 µL dye solution were added to the 96-well plate and incubated for 6 min at 25 °C. The absorbance was measured at 734 nm. The antioxidant activity was calculated using the standard curve of Trolox with concentrations ranging from 0 to 500 µg/mL. Results were expressed in mg of Trolox equivalent per gram of extract (mg TE/g).

### 2.7. Short Chain Fatty Acids (SCFAs) Production (GC-FID) Analysis

SCFAs production of the colonic digesta (FBH and FBE) was assessed according to the protocol described by Gu, Suleria, Dunshea and Howell [21]. Briefly, the acidified post-fermented faeces samples were extracted with water/formic acid (99:1, *v*/*v*) and analysed using gas chromatography (7890B Agilent, Santa Clara, CA, USA) equipped with a flame ionisation detector (FID) using capillary column (SGE BP21, 12 × 0.53 mm internal diameter (ID) with 0.5 μm film thickness, SGE International, Ringwood, VIC, Australia, P/N 054473). The injection volume was 1 μL and 4-methyl-valeric acid was used as the internal standard. Concentration of detected acetic, propionic, and butyric acids were expressed as µmol/L.

### 2.8. Statistical Analysis

Results are presented in means ± standard deviations. All the experiments were performed in triplicates. Experimental data was statistically analysed using one-way ANOVA on Minitab 19 (Minitab^®^ for Windows Release 19, Minitab Inc, Chicago, IL, USA). Tukey’s HSD was calculated to test the significant difference among samples at the level of *p* ≤ 0.05, at a 5 % level of significance.

## 3. Results and Discussions

### 3.1. Changes in the Phenolic Contents and Antioxidant Properties during FBPC Preparation

The effects of concentrating the faba bean protein on the phenolic estimations and antioxidant properties are shown in Table 1. The total phenolic and flavonoid contents of FBPC were 2.79 mg GAE/g and 394.37 µg QE/g, respectively, which were significantly higher than those of FB. Meanwhile, no significant difference was observed in total condensed tannin after making protein concentrate. That might be due to the initial alkaline conditions during production of the concentrate, which could promote the oxidation of condensed tannins [27]. FBPC exhibited significantly higher values in all three antioxidant assays compared with FB. The increased DPPH radical scavenging capacity after processing faba bean into powders with high protein concentration is in agreement with the study of Nnamezie et al. [28] which also showed improved free radical scavenging abilities resulting from greater phenolics content in the extracts with higher concentrations. The higher the phenolic compound concentrations are, the more available hydroxyl groups within the reaction media and, therefore, the more chances for free radicals to donate hydrogen atoms [29].

### 3.2. Effects of Enzymatic Hydrolysis on the Phenolic Contents and Antioxidant Properties

The phenolic estimations and antioxidant potential determinations of all the hydrolysates are shown in Table 1. Total phenolic content and total condensed tannin were significantly enhanced after enzymatic hydrolysis for 30 min. Puspita et al. [30] also reported an increase in the total phenolic content of Alcalase extracts of Japanese wireweed (*Sargassum muticum*). Similar improved phenolic contents in the enzymatic extracts of various fruits (kiwi, pear, green apple, raspberry, blackberry, strawberry and blueberry) and vegetables (pumpkin, green and red pepper) were also observed by Álvarez et al. [31]. At the same time, total flavonoids slightly increased during the first 5 min of hydrolysis but were reduced after longer periods of hydrolysis. This reduction is likely due to anthocyanins, common flavonoidsin faba beans, being highly unstable under alkaline conditions of the Alcalase treatment, and they are decolourised within a short time due to hydration at the 2-position of the anthocyanidin skeleton [32,33]. Ferric reducing antioxidant power of HA_15_, the one subject to hydrolysis for 15 min, was the highest, up to 269.37 µg TE/g, significantly larger than that of the non-hydrolysed one (HA_0_). Interestingly, after 15 min hydrolysis, the FRAP value sharply declined to 197.51 µg TE/g. The increase in FRAP may be ascribed to the breakdown of the peptide bonds and consequent higher hydrogen ion availability, which helps proton donation occur at particular side-chain groups [34]. Shahi et al. [35] added that the enhanced ferric reducing power could result from the cleaved peptide chains releasing amino acids like lysine and tryptophan with antioxidant activities. The strongest DPPH radical scavenging ability was observed in HA_30_, treated with Alcalase for 30 min, up to 0.89 mg TE/g, which was significantly differed from other hydrolysates. Li et al. [36] found that the Alcalase hydrolysates of *Camellia oleifera* seed cake protein showed greater DPPH than papain and trypsin hydrolysates. Similarly, HA_30_ also represented the highest ABTS value (5.36 mg TE/g) but was not significantly differed from the non-hydrolysed one. At the same time, Ali [2] observed a significant increase in the ABTS values of faba hydrolysates after treating with pepsin for 180 min.

Sbroggio, Montilha, FIGUEIREDO, Georgetti and Kurozawa [34] mentioned that the differences between DPPH and ABTS free radical scavenging modes might be related to the peptides produced having different structures. Extensive hydrolysis could result in forming shorter peptides, including dipeptides and tripeptides, and free amino acids, and therefore peptides got higher hydrophilic properties, easier to approach the ABTS free radical that is water-soluble [37]. Chen et al. [38] also confirmed that soybean peptides, due to their hydrophilicity, have difficulties in having interactions with hydrophobic peroxy radicals like DPPH. This could be a reason why the ABTS value was much higher than that of DPPH. Besides, Samaei et al. [39] added that the ABTS assay has higher sensitivity than the DPPH method since the ABTS radical has preferential interactions with the hydroxylated aromatic compounds within the sequences of peptides, which is associated with the protein composition and hydrophobic properties of the hydrolysates, leading to the peptides of hydrolysates having differences in scavenging ABTS and DPPH radicals.

The effect of hydrolysis time on the DPPH was correlated with degree of hydrolysis as shown in SDS-PAGE (Figure 2). According to SDS-PAGE patterns of hydrolysates viz., A₅ to A₃₀ showed that the Alcalase effectively cleaved proteins in hydrolysates and size was significantly reduced with the increase in reaction time. During the initial hydrolysis stage, both enzyme activity and substrate concentration were comparatively higher, resulting in the breakdown of peptide bonds and proteolysis occurring at higher rates [13]. This may lead to the initial increase in DPPH free radical scavenging capacity, and then further interactions of peptides or functional groups of amino acids may reduce DPPH values. Vasconcellos et al. [40] pointed out that glycinin peptides displayed higher antioxidant effects than conglycinin peptides emphasising the importance of protein structure and amino acid composition at constant degree of hydrolysis. At the end of the hydrolysis, A₃₀ resulted in bands under 20 kDa corresponding to glycinin, and that may cause the improved DPPH at A_30._

The results shown in Table 1 revealed that the enzymatic hydrolysis using Alcalase for 30 min is the most effective one, which significantly increased the total phenolic and condensed tannin content and retained or improved DPPH and ABTS radical scavenging capacities but decreased the total flavonoid content and FRAP. The effects of enzymatic hydrolysis were also greatly influenced by the hydrolysis treatment time. Generally, prolonging the hydrolysis time could increase the phenolic content and antioxidant activities of all the hydrolysates to some extent.

### 3.3. Effects of UHT Processing on the Phenolic Content and Antioxidant Properties of Faba Bean Concentrate

The phenolic estimations and antioxidant activities of HA_0_, the slurry of FBPC with adjusted pH and temperature treatment without Alcalase treatment, and EA_0_, the hydrolysate A_0_ with UHT treatment, are shown in Table 1. By comparing HA_0_ and emulsion EA_0_, no significant difference was observed in total phenolic content. However, Xu and Chang [41] found reduced TPC values in yellow and black soybean milk after UHT processing, and Dai et al. [42] reported that the TPC values of UHT-treated strawberry samples were significantly lower than the raw ones. Meanwhile, the total flavonoid content of EA_0_ was noticeably lower than that of HA_0_. Anthocyanins would degrade into various degradation products due to their thermal instability, particularly in a neutral environment [32]. That might be a reason why TFC declined after UHT treatment. While the improved TFC value of UHT-treated yellow soybean milk was attributed to the releasement of free phenolic compounds from lignin within cell walls [43]. It was observed that HA_0_ exhibited a significantly higher total condensed tannin content (0.21 mg CE/g) compared with EA_0_ (0.09 mg CE/g), which indicated UHT greatly decreased? the content of condensed tannins.

In terms of DPPH and ABTS values, HA_0_ and EA_0_ did not significantly differ, revealing that UHT processing retained both DPPH and ABTS radical scavenging abilities. Xu and Chang [41] reported that the DPPH value of UHT soymilk produced from soybeans of Proto variety was considerably higher than the raw soymilk. Besides, heat treatments like cooking and pasteurization may lead to the breakdown of large polymerized structural substances present in the cell wall and the releasement of antioxidant compounds, increasing the antioxidant activity [44]. The FRAP value noticeably decreased from 0.24 µg TE/g (HA_0_) to 0.14 µg TE/g (EA_0_) after UHT processing, resulting from losing natural antioxidants during the thermal processing. However, Nicoli et al. [45] stated that heat treatments might induce compounds having new antioxidant characteristics to form. Dias et al. [46] added that thermal processing would not only result in phenolic compound degradation but also unfold and form heat-induced antioxidants like melanoidins that have great antioxidant potential. These studies agree with the retained DPPH and ABTS values after the UHT processing in this research.

### 3.4. Effects of Enzymatic Hydrolysis Combined with UHT Processing on the Phenolic Contents and Antioxidant Properties

The results shown in Table 1 indicated that the enzymatic hydrolysis using Alcalase for 30 min combined with UHT processing is the most effective one. There was a considerable increase in total phenolic content after this combination processing. The total flavonoid content of the emulsion treated with 30 min hydrolysis and UHT processing showed a significant improvement. Similarly, total condensed tannin noticeably increased to 0.29 mg CE/g after the combination treatment.

Meanwhile, there was no significant difference in both DPPH and ABTS radical scavenging abilities observed after enzymatic hydrolysis for 30 min following the UHT processing. Likewise, ferric reducing power did not show any remarkable change after the combination processing. These indicated that combing enzymatic hydrolysis using Alcalase for 30 min with UHT treatment could retain all the antioxidant activities, while a significant increase in the DPPH values of soy protein isolate hydrolysates, treated with a combination of high-temperature treatment under 121 °C for 10 min and enzymatic hydrolysis using Protamex, was found by Yoo and Chang [47]. Similarly, Voss et al. [48] reported that the okara hydrolysates produced by autoclave processing at 121 °C for 15 min and Alcalase hydrolysis showed a great increase in ABTS value.

### 3.5. Molecular Weight Distribution and Physicochemical Properties of Hydrolysates

Partial enzymatic hydrolysis caused significant changes in the structure of FBPC revealed by SDS-PAGE profiles (Figure 2). C (native FBPC) exhibited four major bands with molecular weights of ~50, ~37, ~30, and ~21 kDa. These bands correspond to the α-subunits, β-subunits and their intermediate subunits of 11S (legumin-like) globulins [49]. The pattern A₀ showed was similar to C but much clearer, revealing that temperature without enzyme had no effect on protein patterns. SDS-PAGE patterns of FBPC hydrolysates viz., A₅ to A₃₀ showed that the Alcalase effectively cleaved proteins in FBPC and size was significantly reduced with the increase in reaction time. A₅ to A₁₅ remarkably reduced the four distinctive bands and was accompanied by the increased occurrence of protein bands whose molecular weight was less than 20 kDa. Further hydrolysis viz., A₃₀ resulted in bands under 15 kDa. SDS-PAGE results agree with the degree of hydrolysis (Table 2). OPA spectrophotometric assay showed increased amounts of α-amino groups with the increased reaction time of Alcalase resulting in ~1% for A₅, ~2% for A₁₀, ~9% for A₁₅, and ~16% for A₃₀.

The physicochemical properties of hydrolysates were also assessed (Table 2). Results showed that prolonging the Alcalase hydrolysis periods increased the solubility of FBPC by 7–23%, with the highest solubility shown in A_3_₀ at pH 7.0. This improved protein solubility might be attributed to the smaller peptides produced by prolonged hydrolysis. These smaller peptides possibly became more soluble due to their formation of stronger hydrogen bonds with water [50]. The electronegativity of FBPC was also enhanced by prolonged hydrolysis at pH 7.0, possibly due to smaller peptides exposed and increased numbers of ionizable amino acids [51]. Besides, Akbari et al. [52] pointed out that enzymatic hydrolysis could dissociate carboxylic groups and release carboxylate ions (COOˉ), resulting in increased electronegativity. In addition, the net negative charge increased with the increase of hydrolysis, which in turn improved protein solubility because of the larger intermolecular repulsive electrostatic forces [5]. Interestingly, the increase in Alcalase hydrolysis also led to a noticeable decrease in surface hydrophobicity (S_o_). The hydrophobic areas were enzymatic cleaved, and the hydrophobic residues exposed were refolded, which might be why S_0_ was reduced [53]. Overall, enzymatic hydrolysis by Alcalase with increased time generally resulted in hydrolysates having better functional properties.

### 3.6. Physicochemical Properties of Emulsions

The physicochemical properties of emulsions were assessed (Table 2). The droplet size of EA_0_ was larger than that of EC, which indicated that UHT processing could increase the particle size. [54] stated that UHT treatment at 135 °C would cause heat-induced protein aggregations on the droplet surface, leading to increased particle size. Other globules like proteins also denature 135 °C. Therefore, in addition to the fat globules, the particle size analysis may include several protein aggregates and protein-fat aggregates as well, resulting in increased particle size [55]. However, the particle size showed a significant reduction after UHT operation with increased Alcalase hydrolysis (EA_5_ to EA_30_). This may be attributed to the smaller peptides with greater emulsifying properties produced by prolonged hydrolysis. With the increase in Alcalase hydrolysis, peptides would have higher electronegativity, protein solubility and lower surface hydrophobicity which helps avoid the aggregation and facilitate small particles to form [16]. Besides, the polydispersity index [56], non-uniformity degree of distribution, was also calculated, and similar trends were observed in PDI, prolonging the Alcalase hydrolysis decreased PDI values.

ζ-potential is measured for predicting the stability of the emulsion. Generally, emulsions with low electronegativity are unstable since storage easily makes them flocculate or coagulate. In contrast, emulsions with higher electronegativity are relatively stable owing to their less attractive forces compared to repulsive forces [57]. The ζ-potential value of EA_0_ (−21.47 mV) was less negative than that of EC (−23.10 mV), which revealed that UHT processing led to lower electronegativity. Protein aggregates would form primarily at high temperatures, leading to decreased protein solubility, which may account for lower electronegativity [58]. With the increase in Alcalase hydrolysis, the number of smaller peptides with more negative charge increased and that might result in the improvement in electronegativity (EA_5_ to EA_30_). The enhanced surface hydrophobicity after UHT processing would also be attributed to the aggregated protein since attractive hydrophobic interactions occur between them, forming the oil droplets coated by protein [57]. Prolonging the hydrolysis caused a significant decrease in the hydrophobicity of UHT treated emulsions, which is consistent with Zang et al. [53]. Overall, emulsions treated with Alcalase hydrolysis were stable after UHT treatment.

### 3.7. Phenolic Estimations across In Vitro Digestion

The phenolic estimations of FBH and FBE across three digestion phases are shown in Table 3. FBE showed no significant difference with FBH in total phenolic content after in vitro digestion. The TPC values of all the samples significantly increased across in vitro digestion. This observation is also consistent with the study of Ribeiro et al. [59] that the total phenolic content of juçara-based smoothies treated with pasteurization and sonication both had a noticeable increase after intestinal digestion. However, Ma et al. [60] reported an increase of 1.64% and a reduction of 19.97% in the TPC values of digested bamboo leaves soup at gastric and intestinal phases, respectively.

The enzymes in the intestinal phase would have actions on the food residual substrate, which helps phenolics release and thus enhance the total phenolic content [61]. Besides, gallic or *p*-coumaric acid and quercetin present in faba beans may increase during digestion since they may convert into other compounds [62]. Zeng et al. [63] added that the phenolics in faba beans are primarily present in their covalent bound form, which may be why they successfully survive the gastric and intestinal digestion to arrive at the colon.

In terms of total flavonoid content, FBE exhibited great total flavonoid contents after intestinal digestion, ranging from 0.80 to 2.61 (mg QE/g) but with almost no detection in the other two phases. Flavonoids are marginally more sensitive to gastric digestion as there were losses observed in all the anthocyanins and the majority of flavonols during the pepsin treatment, particularly higher related to the anthocyanins [64]. This may be responsible for no detection of flavonoids in the gastric phase for all FBE and FBH. On the contrary, Maduwanthi and Marapana [65] found that the gastric digestion stage displayed significantly higher total flavonoid content in bananas (*Musa acuminata*, AAB) treated with ripening agents ethephon and acetylene than the intestinal phase did. The step-wise release of total flavonoids in digests of four different apple varieties was shown from the gastric phase to the intestinal phase in the study of Bouayed et al. [66].

Regarding TCT, the total condensed tannin of some FBH and FBE sharply decreased after oral digestion and even no detection after gastrointestinal digestion. The interactions between tannins and pancreatic enzymes along with the slightly alkaline pH at the intestinal digestion could result in the degradation of tannins [67,68]. Besides, Wojtunik-Kulesza et al. [61] demonstrated that tannins as high molecular weight phenolic compounds could have strong interaction with proteins and get precipitated by the hydrogen bonding and hydrophobic effect. The total condensed tannin of this research was analysed through the filtered supernatant of the digests, which would also be responsible for no detection of TCT for most samples after gastrointestinal digestion.

### 3.8. Antioxidant Activities across the Stages of In Vitro Digestion

The antioxidant activities of FBH and FBE across three digestion phases are shown in Table 3. FBH and FBE performed the highest DPPH in the gastric phase and remarkably decreased after intestinal digestion. It is agreed with the previous studies [59,60,66]. Generally, phenolic compounds like quercetin and caffeic acid are highly stable under an acidic pH environment, like the pH of the gastric phase [69]. However, the intestinal condition with slightly alkaline pH would induce chemical conversions, forming quinone intermediates and other oxidizing compounds which were unstable [70]. Additionally, impair the structures of aglycones, changing the chemical characteristics of antioxidants and thus decreasing their antioxidant potential [71]. Chen et al. [72] also mentioned that the alkaline pH of the pancreatic digestion stage would degrade the antioxidant compounds and thereby causing a reduction in antioxidant activities.

The gastric phase of both FBH and FBE exhibited the lowest FRAP values among the three phases, and this might result from the peptides released in the gastric digestion having limited biological activity [73]. Besides, FBE showed significant higher ferric reducing power compared with FBH across in vitro digestion. This could be attributed to the generation of a brown polymer called melanoidins through the Maillard Reaction and other antioxidative products when emulsions are subjected to UHT treatment at 135 °C [15]. Additionally, the ferric reducing power of FBH and FBE was highest in the intestinal phase. It is consistent with the findings of Chen et al. [74] that five varieties of sesame seeds owned the strongest ferric reducing power at the intestinal phase. However, the DPPH values of bamboo leave soup were comparatively higher at the gastric phase and decreased after intestinal digestion [60].

ABTS values of all the samples significantly improved after gastrointestinal digestion. The ABTS radical scavenging ability was primarily released at the intestinal digestion stage. Changing the environment from acidic to alkaline would result in the deprotonation of the hydroxyl groups on the phenolic aromatic rings and thus improve its antioxidant potential [75]. This supports the study of Bouayed, Hoffmann and Bohn [66] that aglycone form exhibits stronger antioxidant activities than that of the glycoside. The bioactive peptides originating from pancreatin hydrolysis with antioxidant characteristics would also contribute to the improved ABTS radical scavenging ability at the intestinal phase [71]. The increasing tendency across in vitro digestion is agreed by Chen, Lin, Lin, Zheng and Chen [74] and Chotphruethipong et al. [76]. However, a considerable reduction was observed in ABTS values of the red grape pomace after intestinal digestion.

### 3.9. Phenolic Estimations across Colonic Fermentation

The variations of total phenolic compounds, flavonoids, and tannins content in faba bean hydrolysates (FBH) and faba bean emulsions (FBE) during colonic fermentation are displayed in Table 4. It was noted that, during colonic fermentation, total phenolic compounds significantly increased at the first 2 h, except for HA_30_. Nearly all the TPC values at 8 h of fermentation were the highest among the FBH and FBE digesta, while HA_10_ showed the highest TPC value at 4.67 mg GAE/g. However, for EA_30_, it gained its peak at 2 h (3.80 mg GAE/g). Besides, TPC values significantly decreased after 8 h and then got the lowest levels at 24 h of fecal reaction. In terms of the total flavonoids, all FBH and FBE samples illustrated nearly no flavonoids at 0 h. TFC values gradually increased after 2 h of fermentation and significantly dropped after 4 h, except for EA_5_. The emulsions with 5 min enzymatic hydrolysis significantly increased from 4 h, peaked at 8 h (2.61 mg QE/g), and later reduced and gained no detection at 16 h fermentation. As for the total tannin content, HA_5_ was detected with the highest TCT value at 0 h (5.44 mg CE/g), but most FBH and FBE were found to have nearly 0 tannins within the first 8 h of fermentation. All FBH and FBE significantly rose after 8 h, especially for the FBE digesta, getting the highest values at 24 h of fecal reaction.

These clearly demonstrated that phenolic compound release was enhanced throughout colonic fermentation, commonly after 8 h of fermentation, consistent with the study of Wang et al. [77]. Condensed tannins can be found in the plant in the form of oligomers as well as polymers of flavanols, which corresponds to extractable tannins as well as non-extractable tannins. According to Quatrin et al. [78], a large proportion of hydrolysable tannins can be converted quickly by gut microbiota within two hours. Meanwhile, tannins with non-extractable properties that are tightly combined with the protein or around sixty percent of the existing dietary fiber are hard to extract with aqueous acidic, organic solvents or digestive enzymes, however, as they approach the colon, they are metabolized by the gut flora and consequently contribute to the TPC results [79].

Flavonoids frequently form glycosidic bonds with other components. The rapid decrease in TFC values after 4 h of fermentation may be due to the combination of flavonoids with sulphates, glycosides and glucuronides after digestion in the upper digestive tract which are subsequently metabolized in the colon with the help of enzymes such as sulfatase, bacterial glycosidases and glucuronidases [80,81]. During gastrointestinal digestion, only a tiny portion of flavonoids (2–15%) was able to be completely digested and absorbed [80]. The flavonoid aglycon may continuously be metabolized in the presence of gut microbiota, resulting in fission products such as valerolactones and phenolic acids [82]. The degradation of condensed tannins contributes to a gradual rise in colonic TPC value [78]. However, because of some assay limitations, alterations shown in the content of condensed tannins during colonic fermentation are not able to be identified using an ultraviolet spectrophotometer. One of the possible explanations might be interactions among tannin metabolites, other compounds existing in the fermented residue, along with the chemicals applied in the assay, making the precipitate and finally falling within the limits of low determination [83].

### 3.10. Antioxidant Activities across Colonic Fermentation

The variations in the antioxidant potential of faba beans during colonic fermentation were measured by DPPH, FRAP, and ABTS assays, as is displayed in Table 4. In general, a similar tendency in DPPH was found for the digested FBH and FBE, with the values rising and obtaining the highest level after 24 h of fermentation with fluctuations at 4 h and 8 h for some digesta. HA_5_ was observed to have a comparatively higher DPPH value of 8.48 mg TE/g after 24 h, followed by HA_30_ (8.14 mg TE/g). The reducing power of FBH and FBE could be observed through FRAP. FBH and FBE showed an increasing reducing power trend within the first 4 h. In contrast, a decreased trend was noted after 4 h but subsequently increased or sustained from 8 h of fecal reaction. HA_10_ exhibited the highest reducing power after 4 h fermentation with a value of 4.79 mg TE/g. According to the ABTS results, the values of both FBH and FBE samples significantly reduced from 0 h, fluctuated between 2–8 h, but increased considerably after 8 h, and gained the highest ABTS values at 24 h of reaction (123.95 mg TE/g for HA_30_). In addition, it seems that FBH demonstrated a relatively higher trend of ABTS values than the FBE group.

Cárdenas-Castro et al. [84] noted similar alterations in the DPPH value in tomatoes during colonic fermentation, and the DPPH value continued to build up after 24 h of fermentation, which is consistent with our results. This trend maintained upward tendency and might result from the existence of other bioactive compounds in faba beans that express antioxidant potential, such as vitamins [85,86]. Furthermore, the rising trends of antioxidant potential (DPPH and ABTS) that existed throughout colonic fermentation were considerably persistent with the improvement in total phenolic content. It was demonstrated that colonic fermentation could aid in releasing and breaking down phenolics from the residual of FBH and FBE digesta, while also enhancing its in vitro antioxidant potential.

Besides, melanoidins and dietary fiber in faba beans may cause microorganisms in the large intestine to produce compounds such as metabolites of tannins along with phenolic acids, enhancing DPPH, FRAP along with ABTS results [85]. Phenolic acids existing in faba beans are correlated with the in vitro expression of antioxidant capacity [87]. The modifications in antioxidant activities of faba beans at various stages of fermentation also implied that enzymes in the gastrointestinal tract could not completely neutralize biologically active molecules with antioxidant capacity, thus allowing the bioactive substances to enter the colonic site and release their bioactive effects [86,88,89].

### 3.11. Short Chain Fatty Acids Production

Five types of short chain fatty acids, including acetic, propionic, iso-butyric, butyric and valeric acids in FBH and FBE were measured and displayed in Figure 3. The significant difference analysis result of Total SCFAs has been updated in the Supplemetary Materials (Appendix A). Polysaccharides, oligosaccharides, and dietary fibers, containing pectin, beta-glucan, cellulose, gum, etc., will not be digested and absorbed in the upper digestive tract and subsequently, enter the colon. The flora present in the colon will then combine with them for further metabolism. Generally, there was a much lower SCFAs level in our fermented FBH and FBE samples compared to the regular faba bean powder illustrated in another study, in which the level of acetic, butyric and propionic acid after 24 h of reaction were around 41.58 mM, 11.05 mM, and 13.05 mM, respectively [90]. Nonetheless, acetic acid remained the major SCFA produced during colonic fermentation, followed by isobutyric and propionic acids. This finding was consistent with the studies of Gullón, Gullón Estévez, Tavaria, Vasconcelos and Gomes [90] and Wu, Liu, Lu, Barrow, Dunshea and Suleria [89]. However, regarding Çalışkantürk Karataş et al. [91], who measured the potential of faba bean gastrointestinal digesta to enhance gut microbiota fermentation, it is displayed that faba bean digesta facilitated the generation of short chain fatty acids, primarily acetic acid (56.9 µmol), followed by butyric acid (36.1 µmol), propionic acid (23.9 µmol), and valeric acid (8.8 µmol) per 100 mg residue. These slightly different patterns may be due to their use of anaerobic batch cultures to evaluate the impacts on metabolic products.

As shown in Figure 3, during colonic fermentation, the production of SCFAs was found an increasing trend after 4 h of fermentation and peaked after around 16 h, and then sustained or began decreasing steadily. Nevertheless, as noted by Çalışkantürk Karataş, Günay and Sayar [91], the formation of SCFAs still gradually rose after 12 h and showed a peak at 24 h for the whole faba bean. This slow formation pattern might result from the slow pace of fermentation of dietary fiber via gut flora that was picked from the human model, which existed differences from the present study. Periago et al. [92] also discovered that the generation of SCFAs was linked not only to the physical (electrostatic force) and chemical (hydrogen or ester bond) trapping configuration of dietary fiber but also to the species as well as quantities of microbiota.

In addition, the production of SCFAs among FBH and FBE with distinct periods of Alcalase hydrolysis was analogous. Still, it appeared a difference that the SCFAs level in HA_30_ (enzymatic hydrolysis for 30 min) was generally higher than in other treatments. The lower production and the differences among various FBE along with FBH are most likely correlated to the degradation, Maillard reaction along with caramelization of certain carbohydrates when undergoing UHT treatment, which results in a loss of total polysaccharide contents for fermentation [93,94]. The gut microbiota could efficiently metabolize various Maillard reaction products, inducing the generation of short chain fatty acids. Except for HA_30_, the similar level discovered in the fermented samples could be attributed to a substance named melanoidins. Melanoidins could form short chain fatty acids along with colonic fermentation, which finally makes up for the degradation loss caused by UHT treatment [15,94].

## 4. Conclusions

In conclusion, enzymatic hydrolysis of faba bean proteins by Alcalase typically contributed to hydrolysates with better functional properties such as higher protein solubility, ζ-potential and lower surface hydrophobicity. In addition, emulsions treated with Alcalase were found stable after UHT treatment as indicated by EA_0_ having the larger droplet size (14.73 μm), higher hydrophobicity index (244.59), and lower electronegativity (-21.47 mV). Both phenolic contents and antioxidant activities were significantly increased by about 36% and 54%, respectively, in the faba bean protein concentrate compared to the faba bean flour indicating that these compounds and activities partitioned with the proteins. Enzymatic hydrolysis with Alcalase for 30 min resulted in a comparatively higher total phenolic (0.29 mg GAE/g) and condensed tannin content (0.23 mg CE/g) with higher DPPH and ABTS value viz., 0.89 and 5.36 mg TE/g, respectively, but lower TFC and FRAP value (38.69 μg QE/g and 197.51 μg TE/g). UHT treatment maintained TPC, DPPH and ABTS but reduced the TFC by 30.99 μg QE/g and FRAP by 98.24 μg TE/g. Enzymatic hydrolysis combined with UHT significantly increased the content of total phenolics by 0.06 mg GAE/g and retained all the antioxidant activity values of DPPH, FRAP, and ABTS. The release of phenolic compounds in hydrolysates and UHT emulsions increased after intestinal digestion by 0.44 mg GAE/g and 0.55 mg GAE/g, respectively. Intestinal digestion was the stage when most phenolics were released with the highest antioxidant potential observed. For colonic fermentation, the release of phenolics was enhanced via microbiota present in the gut, commonly after 8 h of fermentation. Furthermore, the SCFA production was dominated by acetic acids, which exhibited significant similar changes among hydrolysates and UHT emulsions. Enzymatic hydrolysis of faba bean protein for 30 min resulted in higher antioxidant capacities and SCFAs, which could be more favorable for gut health. Nonetheless, the biotransformation of phenolic compounds after 48 h of colonic fermentation still needs deeper investigation. Overall, this study showed that enzymic hydrolysis of faba proteins not only improved the colloidal emulsion stability, but also released antioxidant capacity during in vitro digestibility and colonic fermentation. Gut microbiome functionality was also affected by hydrolysis for both proteins and emulsions as indicated by the impact of degree of hydrolysis on short chain fatty acid production. More studies are needed to further elucidate and differentiate the role of phenolics during faba protein processing and digestion stages in comparison to contributions of peptides, amino acids and microelements to digestion rates, antioxidant capacities and colonial short chain fatty acid production.

## Figures and Tables

**Figure 1 nutrients-15-00089-f001:**
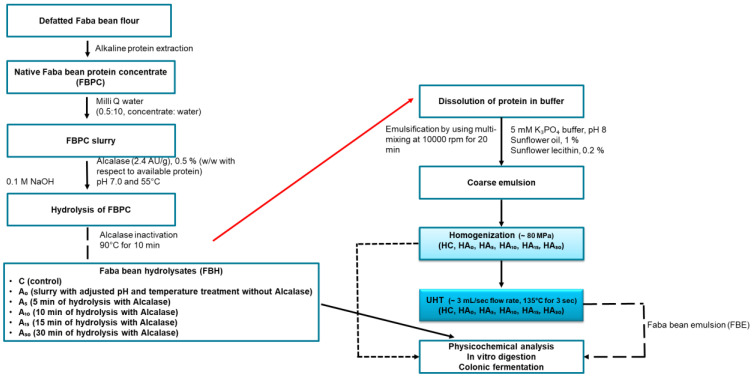
Experimental design.

**Figure 2 nutrients-15-00089-f002:**
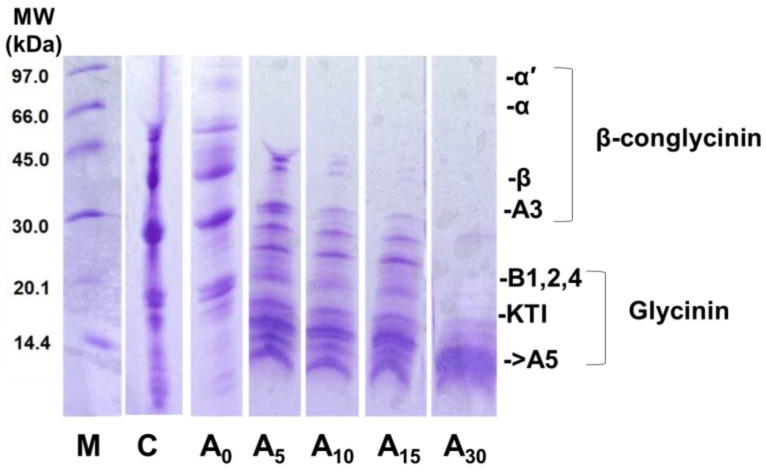
SDS-PAGE profiles of native FBPC protein (C), and FBPC protein hydrolysates viz., slurry with adjusted pH and temperature treatment without Alcalase (A₀), 5 min hydrolysis with Alcalase (A₅), 10 min hydrolysis with Alcalase (A₁₀), 15 min hydrolysis with Alcalase (A₁₅), and 30 min hydrolysis with Alcalase (A₃₀). (M) molecular weight markers.

**Figure 3 nutrients-15-00089-f003:**
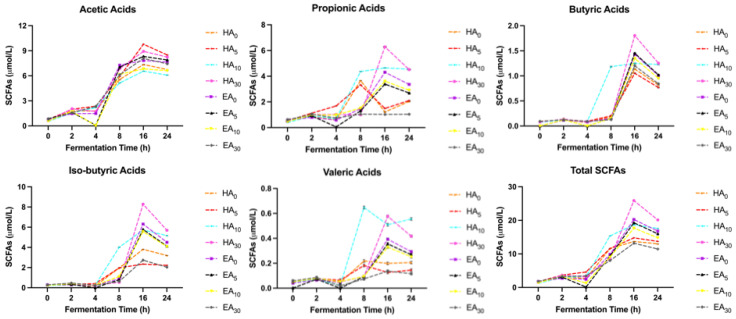
The production of five types of short chain fatty acids, including acetic, propionic, butyric, iso-butyric and valeric acids in FBH (HA_0_, HA_5_, HA_10_, & HA_30_), and FBE (EA_0_, EA_5_, EA_10_, & EA_30_) fermented with six-time variables (0 h, 2 h, 4 h, 8 h, 16 h, & 24 h).

**Table 1 nutrients-15-00089-t001:** The estimations of phenolic content and antioxidant capacities of faba bean (FB), protein concentrate (FBPC), hydrolysates (FBH) and emulsions (FBE) *.

Treatments	Samples	TPC (mg GAE/g)	TFC (µg QE/g)	TCT (mg CE/g)	DPPH (mg TE/g)	FRAP (µg TE/g)	ABTS (mg TE/g)
Faba bean flour	FB	1.78 ± 0.01 ^b^	119.72 ± 13.44 ^b^	1.48 ± 0.18 ^a^	7.18 ± 0.32 ^b^	1650.74 ± 0.01 ^b^	28.06 ± 1.08 ^b^
Faba bean protein concentrate	FBPC	2.79 ± 0.14 ^a^	394.37 ± 28.27 ^a^	1.74 ± 0.05 ^a^	15.58 ± 0.55 ^a^	3271.86 ± 0.19 ^a^	41.50 ± 0.07 ^a^
Hydrolysates (FBH)	HC	0.23 ± 0.01 ^cd^	37.65 ± 0.92 ^c^	0.09 ± 0.01 ^g^	0.86 ± 0.05 ^a^	249.11 ± 3.93 ^b^	5.39 ± 0.61 ^a^
	HA_0_	0.23 ± 0.01 ^d^	50.33 ± 3.98 ^b^	0.09 ± 0.01 ^fg^	0.61 ± 0.02 ^bc^	242.01 ± 9.87 ^b^	4.90 ± 0.53 ^ab^
	HA_5_	0.25 ± 0.01 ^bcd^	63.90 ± 1.13 ^a^	0.14 ± 0.02 ^efg^	0.65 ± 0.01 ^bc^	146.46 ± 2.90 ^de^	3.72 ± 0.25 ^c^
	HA_10_	0.24 ± 0.02 ^cd^	13.71 ± 0.22 ^ef^	0.17 ± 0.02 ^cde^	0.64 ± 0.01 ^bc^	152.61 ± 2.63 ^d^	3.92 ± 0.49 ^bc^
	HA_15_	0.24 ± 0.01 ^cd^	15.45 ± 0.41 ^ef^	0.15 ± 0.02 ^def^	0.51 ± 0.01 ^d^	269.37 ± 2.75 ^a^	4.32 ± 0.25 ^abc^
	HA_30_	0.29 ± 0.01 ^a^	38.69 ± 5.46 ^c^	0.23 ± 0.02 ^b^	0.89 ± 0.08 ^a^	197.51 ± 8.09 ^c^	5.36 ± 0.61 ^a^
Emulsions (FBE)	EC	0.23 ± 0.02 ^d^	30.56 ± 4.15 ^cd^	0.15 ± 0.01 ^de^	0.69 ± 0.02 ^b^	137.63 ± 3.64 ^ef^	4.60 ± 0.37 ^abc^
	EA_0_	0.22 ± 0.00 ^d^	19.34 ± 3.91 ^e^	0.21 ± 0.01 ^bc^	0.65 ± 0.01 ^bc^	143.77 ± 1.88 ^def^	4.48 ± 0.12 ^abc^
	EA_5_	0.25 ± 0.01 ^bcd^	35.12 ± 4.73 ^c^	0.20 ± 0.01 ^bcd^	0.59 ± 0.01 ^c d^	138.67 ± 2.50 ^ef^	4.58 ± 0.15 ^abc^
	EA_10_	0.23 ± 0.02 ^cd^	23.52 ± 2.61 ^de^	0.20 ± 0.01 ^bcd^	0.60 ± 0.01 ^cd^	130.35 ± 1.56 ^f^	4.31 ± 0.26 ^abc^
	EA_15_	0.26 ± 0.01 ^abc^	35.02 ± 3.93 ^c^	0.23 ± 0.02 ^b^	0.58 ± 0.01 ^cd^	135.72 ± 5.90 ^ef^	4.39 ± 0.36 ^abc^
	EA_30_	0.28 ± 0.01 ^ab^	39.15 ± 4.11 ^c^	0.29 ± 0.05 ^a^	0.66 ± 0.03 ^bc^	146.55 ± 2.48 ^de^	4.37 ± 0.24 ^abc^

* Values (mean ± SD) illustrated in this table are in triplicates (n = 3). For each assay, different superscript letters indicated the significant differences among the faba bean (FB) and protein concentrate (FBPC); and hydrolysates (FBH), and emulsions (FBE), respectively, within a column at 95% confidence level (*p* < 0.05). C, native FBPC protein; A₀, FBPC protein hydrolysates viz., slurry with adjusted pH and temperature treatment without Alcalase; A₅, slurry with 5 min hydrolysis with Alcalase; A₁₀, slurry with 10 min hydrolysis with Alcalase; A₁₅, slurry with 15 min hydrolysis with Alcalase; A₃₀, slurry with 30 min hydrolysis with Alcalase; TPC, Total phenolic content; TFC, total flavonoid content; TCT, total condensed tannin; DPPH, 2,2′-diphenyl-1-picrylhydrazyl; FRAP, ferric reducing antioxidant power; ABTS, 2,2′-azinobis-3-ethylbenzo-thiazoline-6-sulfonic acid; GAE, gallic acid equivalents; QE, quercetin equivalents; CE, catechin equivalents; TE, Trolox equivalents.

**Table 2 nutrients-15-00089-t002:** Physicochemical properties (Degree of hydrolysis ~ DH %, Protein Solubility (%), ζ-potential, Hydrophobicity index ~ Sₒ, Droplet size ~ D^3,2^, and Polydispersity index ~ PDI) of faba bean hydrolysates (FBH) and faba bean emulsions (FBE) *.

Treatment	DH %	Protein Solubility (%)	ζ-Potential (mV)	S₀	D^3,2^ (µm)	PDI (%)
HA_0_	0	60.95 ± 0.90 ^a^	−30.93 ± 0.45 ^a^	236.78 ± 3.00 ^a^	-	-
HA_5_	1	67.78 ± 1.57 ^b^	−34.67 ± 0.41 ^b^	205.91 ± 7.46 ^ab^	-	-
HA_10_	2	72.06 ± 0.90 ^c^	−37.53 ± 0.46 ^c^	193.32 ± 2.83 ^ab^	-	-
HA_15_	9	77.14 ± 0.90 ^d^	−39.67 ± 0.36 ^d^	167.23 ± 7.10 ^ab^	-	-
HA_30_	16	83.65 ± 0.67 ^e^	−43.75 ± 0.47 ^e^	122.64 ± 3.52 ^b^	-	-
EC	-	-	−23.10 ± 0.96 ^a^	173.07 ± 1.65 ^b^	14.10 ± 0.46 ^a^	41.55 ± 3.99 ^a^
EA_0_	-	-	−21.47 ± 1.10 ^a^	244.59 ± 13.80 ^d^	14.73 ± 0.35 ^a^	30.16 ± 2.17 ^b^
EA_5_	-	-	−28.45 ± 0.39 ^b^	167.12 ± 5.30 ^b^	0.36 ± 0.06 ^b^	36.99 ± 2.48 ^a^
EA_10_	-	-	−30.65 ± 0.34 ^c^	147.90 ± 2.26 ^a^	0.14 ± 0.01 ^b^	17.73 ± 0.78 ^c^
EA_15_	-	-	−33.56 ± 0.40 ^d^	145.71 ± 1.94 ^a^	0.10 ± 0.01 ^b^	16.31 ± 0.79 ^c^
EA_30_	-	-	−37.60 ± 0.25 ^e^	135.55 ± 2.69 ^a^	0.10 ± 0.01 ^b^	14.31 ± 0.53 ^c^

* Values (mean ± SD) illustrated in this table are in triplicates (n = 3). For each parameter, different superscript letters indicated the significant differences among FBPC protein hydrolysates viz., slurry with adjusted pH and temperature treatment without Alcalase (HA₀), 5 min hydrolysis with Alcalase (HA₅), 10 min hydrolysis with Alcalase (HA₁₀), 15 min hydrolysis with Alcalase (HA₁₅), and 30 min hydrolysis with Alcalase (HA₃₀), and native FBPC protein (EC), FBPC protein hydrolysates viz., slurry with adjusted pH and temperature treatment without Alcalase after UHT treatment (EA₀), 5 min hydrolysis with Alcalase after UHT treatment (EA₅), 10 min hydrolysis with Alcalase after UHT treatment (EA₁₀), 15 min hydrolysis with Alcalase after UHT treatment (EA₁₅), and 30 min hydrolysis with Alcalase after UHT treatment (EA₃₀) within a column at 95% confidence level (*p* < 0.05).

**Table 3 nutrients-15-00089-t003:** Phenolic estimations of faba bean hydrolysates (FBH) and emulsions (FBE) across in vitro digestion *.

Sample Types	Samples	Phases	TPC (mg GAE/g)	TFC (mg QE/g)	TCT (mg CE/g)	DPPH (mg TE/g)	FRAP (mg TE/g)	ABTS (mg TE/g)
Hydrolysates (FBH)	HA_0_	Oral Gastric Intestinal	0.53 ± 0.03 ^c^ 0.59 ± 0.03 ^b^ 1.21 ± 0.15 ^a^	0.02 ± 0.00 ^a^	0.41 ± 0.03 ^a^	0.29 ± 0.02 ^b^	0.43 ± 0.04 ^b^	5.12 ± 0.64 ^b^
				-	-	0.45 ± 0.06 ^a^	0.32 ± 0.03 ^c^	7.61 ± 0.31 ^b^
				-	-	0.13 ± 0.01 ^c^	0.77 ± 0.02 ^a^	66.36 ± 3.82 ^a^
	HA_5_	Oral Gastric Intestinal	0.40 ± 0.01 ^b^ 0.63 ± 0.02 ^a^ 0.39 ± 0.04 ^b^	-	-	0.16 ± 0.02 ^b^	0.41 ± 0.01 ^a^	5.94 ± 0.21 ^c^
				-	-	0.48 ± 0.08 ^a^	0.10 ± 0.01 ^c^	7.90 ± 0.12 ^b^
				-	-	0.07 ± 0.01 ^c^	0.24 ± 0.00 ^b^	22.46 ± 0.39 ^a^
	HA_10_	Oral Gastric Intestinal	0.46 ± 0.04 ^c^ 0.61 ± 0.03 ^b^ 0.92 ± 0.14 ^a^	-	1.16 ± 0.05 ^b^	0.11 ± 0.00 ^b^	0.53 ± 0.03 ^a^	10.52 ± 0.07 ^b^
				-	1.29 ± 0.02 ^a^	0.39 ± 0.05 ^a^	0.10 ± 0.01 ^c^	8.22 ± 0.68 ^c^
				-	-	-	0.50 ± 0.01 ^a^	27.72 ± 0.30 ^a^
	HA_30_	Oral Gastric Intestinal	0.53 ± 0.03 ^c^ 0.79 ± 0.02 ^b^ 1.16 ± 0.11 ^a^	-	0.54 ± 0.08 ^a^	0.37 ± 0.01 ^b^	1.10 ± 0.05 ^a^	12.25 ± 0.09 ^b^
				-	-	0.73 ± 0.06 ^a^	0.45 ± 0.05 ^c^	9.62 ± 0.15 ^b^
				-	-	-	0.58 ± 0.01 ^b^	79.79 ± 0.95 ^a^
Emulsions (FBE)	EA_0_	Oral Gastric Intestinal	0.49 ± 0.03 ^c^ 0.69 ± 0.02 ^b^ 1.15 ± 0.13 ^a^	0.04 ± 0.00 ^b^	2.04 ± 0.02 ^a^	0.28 ± 0.02 ^b^	1.04 ± 0.04 ^b^	12.79 ± 0.41 ^b^
				-	-	0.42 ± 0.05 ^a^	0.90 ± 0.00 ^c^	9.10 ± 0.15 ^b^
				1.40 ± 0.23 ^a^	-	0.19 ± 0.01 ^c^	1.35 ± 0.01 ^a^	116.13 ± 8.91 ^a^
	EA_5_	Oral Gastric Intestinal	0.51 ± 0.02 ^c^ 0.80 ± 0.05 ^b^ 0.65 ± 0.09 ^a^	-	-	0.24 ± 0.02 ^a^	0.99 ± 0.04 ^b^	13.95 ± 0.43 ^b^
				-	-	0.27 ± 0.03 ^a^	0.96 ± 0.07 ^b^	9.94 ± 0.53 ^c^
				0.80 ± 0.11 ^a^	-	0.07 ± 0.01 ^b^	1.10 ± 0.02 ^a^	22.06 ± 0.88 ^a^
	EA_10_	Oral Gastric Intestinal	0.50 ± 0.04 ^c^ 0.79 ± 0.01 ^b^ 1.17 ± 0.08 ^a^	-	2.28 ± 0.02 ^a^	0.29 ± 0.00 ^b^	1.11 ± 0.02 ^b^	14.80 ± 0.76 ^b^
				-	-	0.40 ± 0.02 ^a^	0.93 ± 0.04 ^c^	9.47 ± 0.43 ^b^
				2.61 ± 0.10 ^a^	-	0.14 ± 0.01 ^c^	2.06 ± 0.09 ^a^	78.66 ± 6.38 ^a^
	EA_30_	Oral Gastric Intestinal	0.45 ± 0.04 ^c^ 0.86 ± 0.05 ^b^ 1.17 ± 0.07 ^a^	-	-	0.26 ± 0.02 ^b^	1.00 ± 0.03 ^b^	14.98 ± 0.86 ^b^
				-	-	0.43 ± 0.06 ^a^	1.01 ± 0.05 ^b^	10.00 ± 0.27 ^b^
				0.99 ± 0.14 ^a^	-	0.15 ± 0.01 ^c^	1.22 ± 0.05 ^a^	67.46 ± 1.44 ^a^

* Values (mean ± SD) illustrated in this table are in triplicates (n = 3). The data of all the samples have been subtracted the control values. For each sample of each assay, different superscript letters indicated the significant differences among oral, gastric and intestinal phases within three rows of a column at a 95% confidence level (*p* < 0.05). FBH, faba bean hydrolysates; FBE, faba bean emulsions; A₀, FBPC protein hydrolysates viz., slurry with adjusted pH and temperature treatment without Alcalase; A₅, slurry with 5 min hydrolysis with Alcalase; A₁₀, slurry with 10 min hydrolysis with Alcalase; A₁₅, slurry with 15 min hydrolysis with Alcalase; A₃₀, slurry with 30 min hydrolysis with Alcalase; TPC, Total phenolic content; TFC, total flavonoid content; TCT, total condensed tannin; GAE, gallic acid equivalents; QE, quercetin equivalents; CE, catechin equivalents. DPPH, 2,2′-diphenyl-1-picrylhydrazyl; FRAP, ferric reducing antioxidant power; ABTS, 2,2′-azinobis-3-ethylbenzo-thiazoline-6-sulfonic acid; TE, Trolox equivalent.

**Table 4 nutrients-15-00089-t004:** Phenolic estimations of faba bean (FB), protein concentrate (FBPC), hydrolysates (FBH) and emulsions (FBE) across colonic fermentation *.

Sample Types	Samples	Phases	TPC (mg GAE/g)	TFC (mg QE/g)	TCT (mg CE/g)	DPPH (mg TE/g)	FRAP (mg TE/g)	ABTS (mg TE/g)
Hydrolysates (FBH)	HA_0_	0 h 2 h 4 h	3.39 ± 0.04 ^b^	-	-	2.36 ± 0.01 ^f^	2.91 ± 0.07 ^d^	26.55 ± 1.37 ^d^
			3.44 ± 0.03 ^b^	0.14 ± 0.00 ^c^	-	3.88 ± 0.00 ^d^	3.99 ± 0.02 ^b^	16.15 ± 0.01 ^f^
			2.40 ± 0.07 ^e^	1.06 ± 0.08 ^a^	-	3.45 ± 0.03 ^e^	4.14 ± 0.00 ^a^	33.89 ± 0.12 ^c^
		8 h	3.63 ± 0.06 ^a^	0.45 ± 0.01 ^b^	-	5.78 ± 0.03 ^c^	2.77 ± 0.00 ^e^	19.25 ± 0.00 ^e^
		16 h	2.80 ± 0.00 ^c^	-	-	6.40 ± 0.01 ^b^	3.76 ± 0.00 ^c^	92.52 ± 0.06 ^b^
		24 h	2.52 ± 0.01 ^d^	-	-	6.60 ± 0.03 ^a^	4.09 ± 0.03 ^a^	116.94 ± 0.05 ^a^
	HA_5_	0 h 2 h 4 h	3.17 ± 0.00 ^c^	0.03 ± 0.01 ^d^	5.44 ± 0.13 ^a^	1.13 ± 0.02 ^f^	3.30 ± 0.08 ^d^	33.98 ± 0.35 ^d^
			3.64 ± 0.01 ^b^	0.17 ± 0.00 ^c^	-	4.32 ± 0.00 ^d^	3.45 ± 0.00 ^c^	16.18 ± 0.01 ^f^
			2.94 ± 0.04 ^d^	0.26 ± 0.01 ^b^	-	3.66 ± 0.00 ^e^	4.61 ± 0.01 ^a^	36.17 ± 0.06 ^c^
		8 h	4.07 ± 0.03 ^a^	0.60 ± 0.02 ^a^	-	6.08 ± 0.01 ^c^	1.30 ± 0.00 ^f^	19.35 ± 0.00 ^e^
		16 h	2.52 ± 0.02 ^e^	-	-	7.88 ± 0.26 ^b^	3.05 ± 0.00 ^e^	92.38 ± 0.03 ^b^
		24 h	2.00 ± 0.02 ^f^	-	-	8.48 ± 0.09 ^a^	3.64 ± 0.01 ^b^	116.72 ± 0.06 ^a^
	HA_10_	0 h 2 h 4 h	3.15 ± 0.01 ^c^	-	-	4.27 ± 0.09 ^d^	2.10 ± 0.00 ^f^	23.59 ± 0.37 ^d^
			3.93 ± 0.00 ^b^	0.36 ± 0.00 ^c^	-	5.14 ± 0.01 ^c^	4.53 ± 0.01 ^b^	15.92 ± 0.03 ^f^
			2.81 ± 0.04 ^e^	0.93 ± 0.00 ^b^	-	3.64 ± 0.09 ^e^	4.79 ± 0.00 ^a^	34.52 ± 0.22 ^c^
		8 h	4.67 ± 0.02 ^a^	1.02 ± 0.00 ^a^	-	7.43 ± 0.05 ^b^	3.39 ± 0.00 ^c^	19.16 ± 0.00 ^e^
		16 h	2.95 ± 0.05 ^d^	-	0.91 ± 0.02 ^b^	7.61 ± 0.10 ^a^	3.11 ± 0.01 ^d^	94.60 ± 0.03 ^b^
		24 h	2.38 ± 0.02 ^f^	-	1.22 ± 0.03 ^a^	7.67 ± 0.06 ^a^	3.02 ± 0.01 ^e^	119.75 ± 0.05 ^a^
	HA_30_	0 h 2 h 4 h	3.75 ± 0.00 ^a^	0.20 ± 0.00 ^c^	-	3.11 ± 0.01 ^f^	2.22 ± 0.00 ^e^	56.29 ± 0.26 ^c^
			3.17 ± 0.02 ^d^	0.20 ± 0.00 ^c^	-	5.59 ± 0.00 ^c^	4.26 ± 0.04 ^a^	16.89 ± 0.00 ^f^
			3.15 ± 0.03 ^d^	0.89 ± 0.01 ^a^	-	3.86 ± 0.01 ^e^	4.13 ± 0.00 ^b^	33.69 ± 0.10 ^d^
		8 h	3.70 ± 0.07 ^a^	0.29 ± 0.01 ^b^	-	4.33 ± 0.02 ^d^	3.12 ± 0.00 ^c^	18.99 ± 0.00 ^e^
		16 h	3.43 ± 0.06 ^b^	0.15 ± 0.02 ^d^	0.63 ± 0.01 ^b^	7.19 ± 0.09 ^b^	3.04 ± 0.04 ^d^	97.71 ± 0.04 ^b^
		24 h	3.33 ± 0.03 ^c^	0.11 ± 0.01 ^e^	0.83 ± 0.02 ^a^	8.14 ± 0.04 ^a^	3.01 ± 0.02 ^d^	123.95 ± 0.03 ^a^
Emulsions (FBE)	EA_0_	0 h 2 h 4 h	3.17 ± 0.02 ^d^	-	-	3.00 ± 0.03 ^f^	3.52 ± 0.05 ^c^	19.65 ± 0.05 ^d^
			3.32 ± 0.03 ^c^	0.38 ± 0.00 ^b^	-	4.87 ± 0.05 ^d^	3.72 ± 0.00 ^b^	15.33 ± 0.03 ^f^
			2.45 ± 0.02 ^e^	0.96 ± 0.02 ^a^	-	3.77 ± 0.05 ^e^	4.38 ± 0.00 ^a^	50.07 ± 0.09 ^c^
		8 h	3.85 ± 0.01 ^a^	0.29 ± 0.00 ^c^	-	6.10 ± 0.00 ^c^	2.51 ± 0.00 ^f^	18.69 ± 0.00 ^e^
		16 h	3.63 ± 0.04 ^b^	-	5.51 ± 0.00 ^b^	6.84 ± 0.05 ^b^	3.15 ± 0.06 ^e^	97.17 ± 0.07 ^b^
		24 h	3.56 ± 0.02 ^b^	-	7.35 ± 0.02 ^a^	7.08 ± 0.03 ^a^	3.36 ± 0.04 ^d^	123.33 ± 0.06 ^a^
	EA_5_	0 h 2 h 4 h	2.31 ± 0.01 ^f^	0.18 ± 0.00 ^c^	-	3.31 ± 0.02 ^e^	2.56 ± 0.01 ^e^	61.22 ± 0.01 ^c^
			3.71 ± 0.01 ^b^	0.26 ± 0.00 ^b^	-	5.01 ± 0.00 ^c^	3.57 ± 0.00 ^b^	15.88 ± 0.01 ^e^
			2.83 ± 0.13 ^e^	0.25 ± 0.02 ^b^	-	3.45 ± 0.04 ^de^	4.70 ± 0.00 ^a^	57.03 ± 0.19 ^d^
		8 h	4.20 ± 0.01 ^a^	2.61 ± 0.00 ^a^	-	3.58 ± 0.03 ^d^	2.39 ± 0.00 ^f^	61.11 ± 0.01 ^c^
		16 h	3.25 ± 0.01 ^c^	-	2.35 ± 0.01 ^b^	5.99 ± 0.13 ^b^	3.02 ± 0.01 ^d^	88.35 ± 0.01 ^b^
		24 h	2.93 ± 0.02 ^d^	-	3.13 ± 0.01 ^a^	6.80 ± 0.05 ^a^	3.23 ± 0.01 ^c^	97.43 ± 0.08 ^a^
	EA_10_	0 h 2 h 4 h	3.51 ± 0.00 ^b^	0.10 ± 0.00 ^b^	-	2.23 ± 0.10 ^e^	2.32 ± 0.01 ^f^	61.28 ± 0.02 ^c^
			3.40 ± 0.06 ^c^	0.06 ± 0.00 ^c^	-	6.31 ± 0.02 ^a^	4.09 ± 0.00 ^a^	16.67 ± 0.00 ^f^
			3.55 ± 0.01 ^b^	0.22 ± 0.02 ^a^	-	5.29 ± 0.05 ^d^	3.94 ± 0.00 ^b^	36.02 ± 0.06 ^d^
		8 h	3.93 ± 0.01 ^a^	0.10 ± 0.00 ^b^	-	5.66 ± 0.05 ^c^	2.42 ± 0.00 ^e^	19.33 ± 0.00 ^e^
		16 h	3.43 ± 0.01 ^c^	-	2.20 ± 0.01 ^b^	5.87 ± 0.01 ^b^	2.86 ± 0.03 ^d^	87.43 ± 0.03 ^b^
		24 h	3.26 ± 0.01 ^d^	-	2.93 ± 0.02 ^a^	5.95 ± 0.02 ^b^	3.01 ± 0.02 ^c^	110.13 ± 0.05 ^a^
	EA_30_	0 h 2 h 4 h	2.12 ± 0.02 ^d^	-	0.53 ± 0.01 ^c^	1.85 ± 0.01 ^f^	2.51 ± 0.00 ^e^	76.29 ± 1.26 ^c^
			3.80 ± 0.03 ^a^	0.15 ± 0.00 ^b^	-	5.92 ± 0.00 ^c^	3.41 ± 0.00 ^b^	17.02 ± 0.00 ^f^
			3.19 ± 0.02 ^b^	0.88 ± 0.00 ^a^	-	4.46 ± 0.05 ^e^	4.55 ± 0.01 ^a^	32.61 ± 0.24 ^d^
		8 h	2.90 ± 0.06 ^c^	-	-	4.96 ± 0.02 ^d^	2.41 ± 0.00 ^f^	18.95 ± 0.01 ^e^
		16 h	1.93 ± 0.00 ^e^	-	4.13 ± 0.01 ^b^	7.03 ± 0.16 ^b^	2.70 ± 0.00 ^d^	84.57 ± 0.18 ^b^
		24 h	1.61 ± 0.02 ^f^	-	5.51 ± 0.02 ^a^	7.72 ± 0.06 ^a^	2.79 ± 0.02 ^c^	106.45 ± 0.09 ^a^

* Values (mean ± SD) illustrated in this table are in triplicates (n = 3). The data of all the samples have been subtracted the control values. For each sample of each assay, different superscript letters indicated the significant differences among 0 h, 2 h, 4 h, 8 h, 16 h and 24 h phases within five rows of a column at a 95% confidence level (*p* < 0.05). FBH, faba bean hydrolysates; FBE, faba bean emulsions; A₀, FBPC protein hydrolysates viz., slurry with adjusted pH and temperature treatment without Alcalase; A₅, slurry with 5 min hydrolysis with Alcalase; A₁₀, slurry with 10 min hydrolysis with Alcalase; A₁₅, slurry with 15 min hydrolysis with Alcalase; A₃₀, slurry with 30 min hydrolysis with Alcalase; TPC, Total phenolic content; TFC, total flavonoid content; TCT, total condensed tannin; GAE, gallic acid equivalents; QE, quercetin equivalents; CE, catechin equivalents. DPPH, 2,2′-diphenyl-1-picrylhydrazyl; FRAP, ferric reducing antioxidant power; ABTS, 2,2′-azinobis-3-ethylbenzo-thiazoline-6-sulfonic acid; TE, Trolox equivalent.

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
