# Peer review of "In Vitro Digestion and Colonic Fermentation of UHT Treated Faba Protein Emulsions: Effects of Enzymatic Hydrolysis and Thermal Processing on Proteins and Phenolics"

_nutrients, 2022, doi:10.3390/nu15010089_

Round 1

Reviewer 1 Report

The manuscript “In vitro digestion and colonic fermentation on the bioavailability of phenolic compounds in UHT treated faba protein emulsions: effects of enzymatic hydrolysis and thermal processing on proteins and phenolics”. There are many groups in the manuscript. I did not find a clear purpose of the study, research significance and conclusions. Changes in phenolic substances are key to affecting antioxidant capacity. However, I found no data results on changes in the composition and content of phenolic substances. In addition, the results of antioxidant capacity data do not change regularly. I thought that there were too many groups and the data were not regular, which ultimately led to unclear purpose and inability to draw a firm conclusion. Overall, the study lacked innovation. The analysis method is relatively simple, and there is no most important HPLC data. Through this study, how are the intake patterns of FBE and FBH chosen?

1. Is the operation of UHT necessary in the Emulations preparation experiment?

2. The broad bean phenolics were extracted. Why didn't the composition and content of phenolic substances be measured? What is the role of FBP in this study? I didn't see an analysis of the results for FBP. UHT can lead to the biotransformation of phenolic acids and flavonoids, which may be the key to their antioxidant capacity. Digestion and fermentation can also lead to changes in phenolic composition, for example, in fermentation experiments, will microorganisms use more phenolic acids (the content of gallic acid, chlorogenic acid and ferulic acid was significantly reduced) or more flavonoids (the content of catechins and quercetins was significantly reduced)?

3. Why is human feces not selected? The authors indicate the pig feces was comparable to human feces, but they didn’t show the sequencing data as supplemental data.

4. After simulating in vitro digestion, dialysis and bioaccessibility analysis were not performed.

5. In colonic fermentation experiments, the pH was not measured?

6. The SCFA unit can be changed to nmol/L, and no error value and significance analysis are seen.

7. Changes in TPC and TFC should be positively correlated with changes in antioxidant capacity. However, I found that the lower content of TPC in Table 4 has a higher antioxidant capacity. Of course, at least two methods are required to determine antioxidant capacity, and three methods are used here, but their results are not consistent. This may also be related to the principle of the antioxidant method.

Author Response

We thank the editor and reviewers for their careful reading of the manuscript and their constructive comments. The comments were very useful to improve the clarity of the work presented. We have taken the comments on board to improve and clarify the manuscript. Please find below our point-by-point response to all comments (reviewers’ comments in black, our replies in blue).

Reviewer 1:

There are many groups in the manuscript. I did not find a clear purpose of the study, research significance and conclusions. Changes in phenolic substances are key to affecting antioxidant capacity. However, I found no data results on changes in the composition and content of phenolic substances. In addition, the results of antioxidant capacity data do not change regularly. I thought that there were too many groups, and the data were not regular, which ultimately led to unclear purpose and inability to draw a firm conclusion. Overall, the study lacked innovation. The analysis method is relatively simple, and there is no most important HPLC data. Through this study, how are the intake patterns of FBE and FBH chosen?

Response: We appreciate the reviewer’s comments and suggestions to improve the quality of the manuscript.

  1. To clarify the rationale and hypothesis building around the many groups in the manuscript a pictorial representation of experimental design was added in the materials and methods section of the revised manuscript (Figure 1).
  2. The purpose/objective of the study is outlined in the last paragraph of the introduction. The overall research significance and conclusions are summarised in the conclusion section and form part of the abstract
  • We agree that the antioxidant capacity may be largely determined by the content, structure, and composition of the phenolic substances. However, our systems comprise other entities that might contribute significantly to antioxidant capacity such as amino acids, peptides, microelements, and Maillard Reaction Products.  In order the further elucidate the role of phenolics in antioxidant capacity, we have characterised and are completing interpretation of   findings from LC-ESI-QTOF-MS/MS, HPLC, and LC-ESI- QqQ-MS characterisation. This investigation will form the basis for a future manuscript.  
  1. We have indicated in the last paragraph of the introduction the innovation and science gap as, “the knowledge on how UHT affects the phenolic estimations and antioxidant properties of faba bean protein emulsions is limited. There is a lack of comprehensive understanding of how in vitro gastrointestinal digestion influences the release of phenolic compounds in hydrolysates and emulsions”.
  2. Is the operation of UHT necessary in the Emulations preparation experiment?

Response: Commercial plant-based beverages are largely processed via UHT to extend the shelf stability It is known that this heat treatment transforms the molecular state of milk components through heat induced reactions and we have investigated how this heat treatment will affect the quality attributes of the emulsions (i.e., digestibility, particle size distribution, stability, antioxidant capacity etc).

  1. The broad bean phenolics were extracted. Why didn't the composition and content of phenolic substances be measured? What is the role of FBP in this study? I didn't see an analysis of the results for FBP. UHT can lead to the biotransformation of phenolic acids and flavonoids, which may be the key to their antioxidant capacity. Digestion and fermentation can also lead to changes in phenolic composition, for example, in fermentation experiments, will microorganisms use more phenolic acids (the content of gallic acid, chlorogenic acid and ferulic acid was significantly reduced) or more flavonoids (the content of catechins and quercetins was significantly reduced)?

Response: The estimation results of phenolic content and antioxidant capacities from FBP were used as a reference value for this study. Faba bean is a good source of protein and phenolics, especially condensed tannins. However, the high content of proteins could trap the phenolic compounds and negatively influence the releasement of phenolics during digestion and their related bioactivities (antioxidant activities). Through the comparison of the crude estimation of phenolic content and antioxidant activities from FBP and FBPC, we could have a rough concept of bound phenolics originally presented in faba beans so that a better understanding of different processing impacts. UHT could induce the releasement of partial phenolics from protein and further degradation of existing free phenolics. The remaining bound phenolics are understood to be only scarcely liberated by the digestive enzymes until the action of gut microflora during colonic fermentation, which could change the phenolic content and antioxidant activities.

  1. Why is human feces not selected? The authors indicate the pig feces was comparable to human feces, but they didn’t show the sequencing data as supplemental data.

Response: The authors agree with the reviewer that ideally human faeces should be selected for colonic fermentation studies. However, for human faecal studies there are ethical issues, approval processes and volunteer recruitment involved and for this short baseline study the pig faeces were used as a model. The pig faecal model is widely accepted for in vitro colonic fermentation studies to predict human digestion.  and we have included into the manuscript a comment on the shortcomings of the faecal model used.

  1. After simulating in vitro digestion, dialysis and bioaccessibility analysis were not performed.

Response: This study aimed to understand how the content of overall bound phenolics in different processed faba bean protein concentrates and their antioxidant activities would be changed during in vitro digestion and colonic fermentation. For this pilot study it was more meaningful and efficient to confirm the overall phenolic content and antioxidant activities before evaluating the bioaccessibility and dialysis rate of specific phenolic compounds bounded with protein in the faba bean using the dialysis membrane. We have made several amendments in the manuscript to eliminate the suggestion that we performed bioaccessibilty evaluations by modifying: - the title to “In vitro digestion and colonic fermentation of UHT treated faba protein emulsions: effects of enzymatic hydrolysis and thermal processing on proteins and phenolics”.

  1. In colonic fermentation experiments, the pH was not measured?

Response: Considering this pilot study is for general background understanding rather than a specific investigation of the gut microbiome composition and functionality. we did not continuously monitor pH, buffering capacity and other intrinsic parameter during digestions. However, we selectively checked the pH of digesta after different colonic fermentation intervals which fluctuated generally around pH 6.8.

  1. The SCFA unit can be changed to nmol/L, and no error value and significance analysis are seen.

Response: The SCFA unit has been converted to µmol/L. The error values were also added as error bars in Figure 3. Our purpose was to understand the trend of SFCA production during the colonic fermentation of faba bean hydrolysates (FBH) and emulsion (FBE). So, we chose a line chart to present the overall tendency and differences between FBH and FBE. All experimental data were assessed by one-way analysis of variance (ANOVA) to determine the significant differences (p < 0.05). The significant difference analysis result of Total SCFAs has been updated in the supplementary file as an example.

  1. Changes in TPC and TFC should be positively correlated with changes in antioxidant capacity. However, I found that the lower content of TPC in Table 4 has a higher antioxidant capacity. Of course, at least two methods are required to determine antioxidant capacity, and three methods are used here, but their results are not consistent. This may also be related to the principle of the antioxidant method.

Response: Thanks for your careful reading and the discussions on the usefulness of antioxidant methods in these multicomponent samples. As you suggest the interpretation and correlation of data from these assays is complicated as each method measured a different antioxidant mechanism in our multicomponent systems. i) The DPPH method has the ability of the various antioxidants to donate an electron or hydrogen radical to the stable DPPH free radical, ii)FRAP method compares antioxidants based on their ability to reduce ferric (Fe3+) to ferrous (Fe2+) ion through the donation of an electron, with the resulting ferrous ion (Fe2+), and iii)The ABTS assay is based on the generation of a blue/green ABTS+ that can be reduced by antioxidants, whereas the DPPH assay is based on the reduction of the purple DPPH to 1,1-diphenyl-2-picryl hydrazine. The ABTS assay is more sensitive to the hydrophilic compounds than the DPPH assay, which could result in value differences. Apart from phenolic compounds, other metabolites generated by gut microflora during colonic fermentation could also contribute to the antioxidant potential making interpretation and correlation complex compared with a single component system where single variable correlations can be observed.

Reviewer 2 Report

The manuscript by Gu et al. entitled “In vitro digestion and colonic fermentation on the bioavailability of phenolic compounds in UHT treated faba protein emulsions: effects of enzymatic hydrolysis and thermal processing on proteins and phenolics” concerns evaluation of various physicochemical properties of faba protein emulsions subjected to some processing methods.

The manuscript covers many experimental procedure, it is well designed, well written and interesting for the readers.

Please find below my minor comments:

1. In Abstract line 16, replace „physiochemical” with “physicochemical”

2. In Abstract short conclusion of the performed studies and results should be included.

3. At the end of introduction (lines 66-69) in the sentence: ” Thus, the objective of this study was to determine how physicochemical properties, phenolics, and anti-oxidant activities, changed during in vitro gastrointestinal digestion and colonic fermentation in faba bean hydrolysates and emulsions.” Information of the processing of the faba been.

4. Lines 77-78 : correction required in the name: “2,2’-diphenyl-1-picrylhydrazl”

5. Please indicate the source of Alcalase.

6. In the line 349 please modify the citation.

7. In the list of references some corrections are required in numbers 2 and 34.

Author Response

We thank the editor and reviewers for their careful reading of the manuscript and their constructive comments. The comments were very useful to improve the clarity of the work presented. We have taken the comments on board to improve and clarify the manuscript. Please find below our point-by-point response to all comments (reviewers’ comments in black, our replies in blue).

The manuscript covers many experimental procedure, it is well designed, well written and interesting for the readers.

Response: Thank you for showing interest in our study. and the complementary comments.

Please find below my minor comments:

  1. In Abstract line 16, replace „physiochemical” with “physicochemical”

Response: Corrected.

  1. In Abstract short conclusion of the performed studies and results should be included.

Response: Corrected. Brief conclusion is added in abstract.

  1. At the end of introduction (lines 66-69) in the sentence:” Thus, the objective of this study was to determine how physicochemical properties, phenolics, and anti-oxidant activities, changed during in vitro gastrointestinal digestion and colonic fermentation in faba bean hydrolysates and emulsions.” Information of the processing of the faba been.

Response: Corrected. The statement is revised for better understanding. Revised statement is, “Thus, the objective of this study was to determine how physicochemical properties, phenolics, and antioxidant activities, changed during in vitro gastrointestinal digestion and colonic fermentation in faba bean protein hydrolysates and oil-in-water emulsions emulsified by these faba been protein hydrolysates”.

  1. Lines 77-78 : correction required in the name: “2,2’-diphenyl-1-picrylhydrazl”

Response: Corrected.

  1. Please indicate the source of Alcalase.

Response: Corrected. Source of Alcalase mentioned in the revised manuscript.

  1. In the line 349 please modify the citation.

Response: The citation is correct as it is a solo author research paper. Therefore, only single author name “Ali” is mentioned with bibliography number.

  1. In the list of references some corrections are required in numbers 2 and 34.

Response: Corrected.

Reviewer 3 Report

Work by Gu et al. It is a very interesting piece of literature that is very well written. The introduction section introduces the reader to the issue well and the clearly written materials and methods refer well to the given research. I have only a few editorial comments regarding the increased number of spaces in certain places in the text, moving table 4 to the next page and increasing the size and resolution of figure 2, because the current form is very hard to read.

Author Response

We thank the editor and reviewers for their careful reading of the manuscript and their constructive comments. The comments were very useful to improve the clarity of the work presented. We have taken the comments on board to improve and clarify the manuscript. Please find below our point-by-point response to all comments (reviewers’ comments in black, our replies in blue).

Work by Gu et al. It is a very interesting piece of literature that is very well written. The introduction section introduces the reader to the issue well and the clearly written materials and methods refer well to the given research. I have only a few editorial comments regarding the increased number of spaces in certain places in the text, moving table 4 to the next page and increasing the size and resolution of figure 2, because the current form is very hard to read.

Response: Thank you for showing interest and support for our study. We have rectified the spacing errors; Table 4 is now moved to next page and Figure 2 (figure 3 in revised version) is replaced with high image.

Round 2

Reviewer 1 Report

There is no more questions.